# Incipient sympatric speciation in wild barley caused by geological-edaphic divergence

Kexin Li[1,2,3,*], Xifeng Ren[1,*], Xiaoying Song[2], Xiujuan Li[4], Yu Zhou[1], Eli Harlev[3], Dongfa Sun[1,†], Eviatar Nevo[3,†]

**Sympatric speciation (SS) has been contentious since the idea was suggested by Darwin. Here, we show in wild barley SS due to geologic and edaphic divergence in "Evolution Plateau," Upper Galilee, Israel. Our whole genome resequencing data showed SS separating between the progenitor old Senonian chalk and abutting derivative young Pleistocene basalt wild barley populations. The basalt wild barley species unfolds larger effective population size, lower recombination rates, and larger genetic diversity. Both species populations show similar descending trend ~200,000 yr ago associated with the last glacial maximum. Coalescent demography analysis indicates that SS was local, *primary*, in situ, and not due to a *secondary* contact from ex situ allopatric population. Adaptive divergent putatively selected genes were identified in both populations. Remarkably, disease resistant genes were selected in the wet basalt population, and genes related to flowering time, leading to temporal reproductive isolation, were selected in the chalk population. The evidence substantiates adaptive ecological SS in wild barley, highlighting the genome landscape during SS with gene flow, due to geologic-edaphic divergence.**

## Introduction

Sympatric speciation (SS), the origin of new species in a free breeding population with gene flow (Darwin, 1859), and two contrasting ecologies, remains highly debated and considered *rare* by most biologists (Hendry, 2009). By contrast, *allopatric* speciation in geographically isolated populations, without the homogenizing effects of gene flow, is considered *common*. Notably, however, sharply divergent ecologies, *microclimatically*, as in the Evolution Canyon (EC) model (Nevo, 1995) in Mount Carmel, or geologically and *edaphically* as in the Evolution Plateau (EP) model in the blind subterranean mole rat, *Spalax galili* (Polyakov et al, 2004; Hadid et al, 2013; Li et al, 2015; 2016b), could illustrate the contentious issue

of SS origin, evolution, and abundance. Such cases, identified as *preliminary* origin of SS, with gene flow, and contrasting ecologies, are decisive evidence for *incipient* or *full* SS, once reproductive isolation is revealed. Our first long-studied case of SS at EP has been the blind mole rat, *S. galili*. Was it unique, or, might it suggest that the sharp ecological contrast of chalk-basalt is a new hot spot of SS?

The vegetation at EP, located at eastern Upper Galilee, Israel, is typical Mediterranean (Hadid et al, 2013). Grazing resulted in elimination of live oak, *Quercus calliprinos*, forest replaced by batha plant formation, dominated by *Sarcopoterium* shrublets and annual and perennial herbaceous plants. The abutting geological formations, Senonian chalk, and Pleistocene volcanic basalt at EP (see geological map in Fig 1A) are rich in herbaceous flora but differ in vegetation (Hadid et al, 2013). The more *humid* basalt area is dominated by ephemeral herbaceous flora. By contrast, the more *arid* chalk is covered by *Sarcopoterium* shrublets cover and ephemeral plants, mostly perennials. Species number in the basalt and chalk abutting habitats are 76 and 69, respectively, sharing *only* 32 plant species (28%) (Hadid et al, 2013), revealing dramatic plant biodiversity divergence between the abutting *carbonaceous* chalk and *siliceous* basalt habitats. The basalt is dominated by a combination of more temperate Mediterranean elements (e.g., *Dactylis glomerata*, several *Trifolium* species, and *Acanthus syriacus*), whereas the chalk is dominated by more arid elements (e.g., *Sarcopoterium spinosum*, *Ceratocephala falcata*, and *Teucrium polium*). Notably, two other bulbs diverge between the rendzina and basalt soils, *Crocus hyemalis* in chalk and *Crocus aleppicus* in basalt-indicating that other taxa also speciate *sympatrically* between chalk and basalt (Hadid et al, 2013), as did the blind mole rats, suggesting that the abutting soil divergence is promoting SS.

To test whether the contrasting and abutting chalk and basalt at EP is indeed a hot spot of SS, rather than a simple *local* adaptive divergence, we selected the annual diploid (2n = 14) wild barley, WB, *Hordeum spontaneum*, the progenitor of cultivated barley, as our test case. We have been studying ecologically genetically wild barley across the Near East Fertile, since 1975. The origin and diversity of WB is in the Fertile Crescent region, including the EP microsite, where it is rich in genetic polymorphism, important for

[1]College of Plant Science and Technology, Huazhong Agricultural University, Wuhan, China   [2]State Key Laboratory of Grassland Agro-Ecosystem, Institute of Innovation Ecology, Lanzhou University, Lanzhou, China   [3]Institute of Evolution, University of Haifa, Haifa, Israel   [4]School of Life Sciences, Zhengzhou University, Zhengzhou, China

Correspondence: nevo@evo.haifa.ac.il; sundongfa@mail.hzau.edu.cn
*Kexin Li and Xifeng Ren contributed equally to this work
†Dongfa Sun and Eviatar Nevo contributed equally to this work

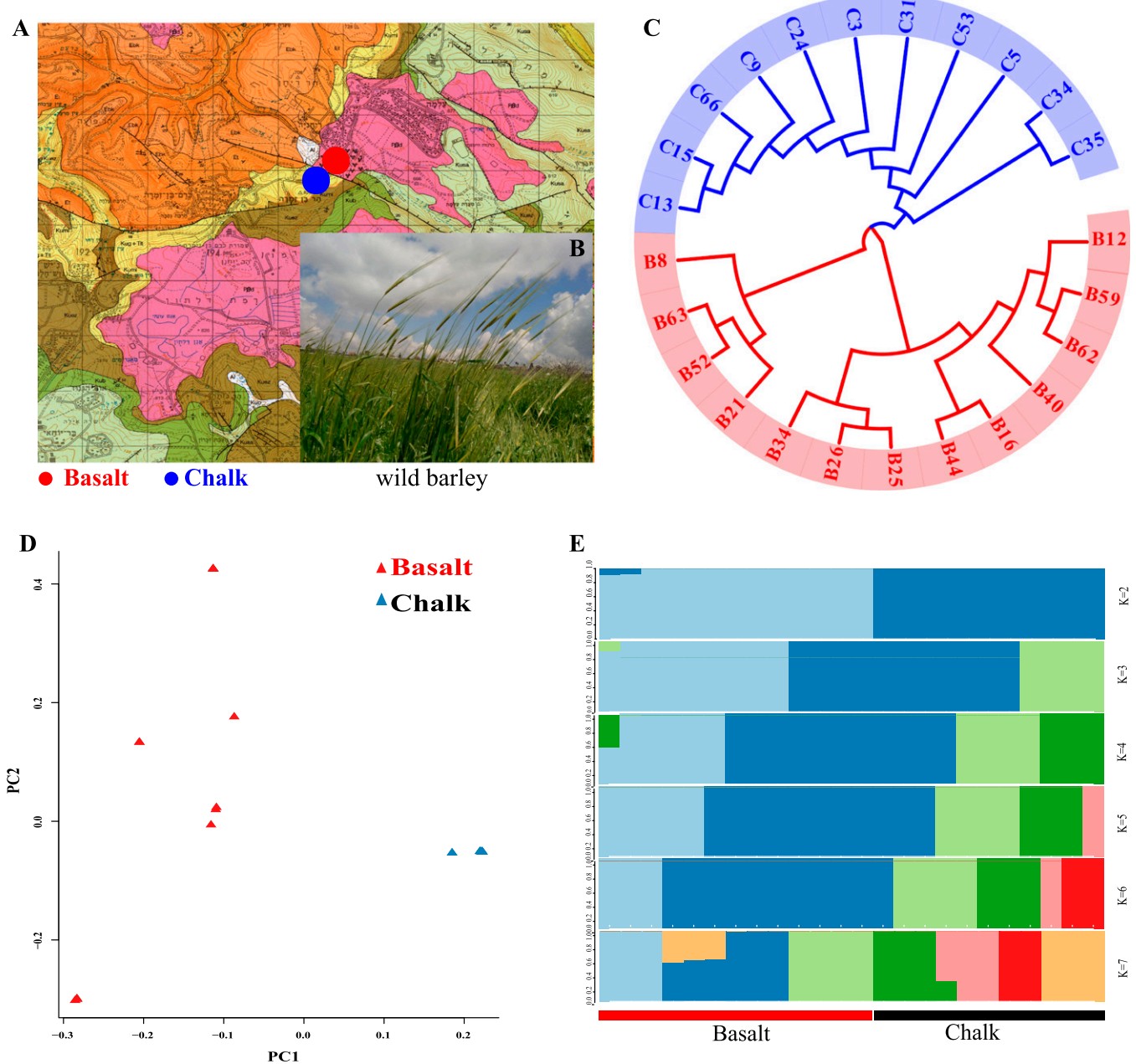

**Figure 1.   Ecological and genetic differentiation of wild barley, *Hordeum spontaneum*, populations from chalk and abutting basalt soils from Evolution Plateau, eastern Upper Galilee, Israel.**
**(A)** Geological map of eastern Upper Galilee (Levitte, 2001). Red denotes the Pleistocene volcanic basalt abutting with the Senonian chalk denoted by yellow. **(B)** The green spiking wild barley before maturation appears at the lower right. Vegetation differentiation of chalk, covered by thick perennial bushlets of *S. spinosum*, and separated sharply from the abutting basalt soil, are revealed by brown mounds of subterranean blind mole rats. Perennial plants include *Dactylis glomeratta*, and *Acanthus syriaca*, plus annuals including several species of *Trifolium* (see details in Fig S5 [Hadid et al, 2013]). Plants were identified by Ori Fragman-Sapir. **(C)** Neighbor-joining tree of wild barley from Basalt (13 individuals, marked in red) and abutting Chalk (11 individuals, marked in blue). **(D)** Principal component analysis of the two soil wild barley populations, basalt population was marked by red triangles, whereas the chalk population was marked by blue triangles. **(E)** Population genetic structure of wild barley from chalk and basalt soil populations, and genetic cluster (k) was set from two to seven, respectively. Color stands for different ancestry constitute.

testing SS. HS is a generalist, hardy wild cereal, growing in multiple soil types, and can even penetrate the central Negev Desert till Mitzpe Ramon (Nevo, 2013). Moreover, we have shown earlier that HS undergoes incipient SS in EC because of interslope *microclimatic* contrasts (Nevo, 2006; Parnas, 2006). and a parallel genome

study on SS of WB at EC is currently under review. To our best knowledge, our studies on SS of WB are the first ones across its distribution range.

Our studies at the EC I focused on adaptive evolution and incipient or full SS across life. In both studies of SS in EC I and EP, we

asked will WB also undergo SS at EC I, like other organisms from bacteria to mammals, due to *microclimatic* interslope divergence (Nevo, 2014), and at EP due to *geologic-edaphic contrast of chalk versus basalt*, as we showed recently at EP microsite in subterranean blind mole rats, *S. galili* (Hadid et al, 2013; Li et al, 2015, 2016b)? Our results unfolded clearly preliminary SS of wild barley at both EC I and at EP. Phenotypic studies at both microsites will follow in future studies.

# Results

### Studied populations of wild barley and whole genome resequencing

The WB population samples were collected from chalk and its abutting basalt soils (Figs 1A and S1A) on a 100-m transect on each soil, and 1–2 m between the collected WB plants (Fig 1B). Seedling leaves were harvested for whole genome sequencing (WGS). After WGS, we got 5.53T clean data, and the data quantity and quality were listed in Table S1. The average sequencing depth is 15.2× and total sequence depth is 364× (Table S2). The number of clean reads, mapped reads, mapping rate, average depth of each individual from both chalk, and abutting basalt are listed in Table S2. We identified 14,490,519 single nucleotide polymorphism (SNP) in total, in which 12,512,900 were *unique* to chalk population (Fig S1B). By contrast, *only* 1,973,396 were unique to basalt population. *Only* 42,238 SNPs (Fig S1B) were shared between the two wild barley soil populations (Fig 1B). These dramatic SNP differences and the low number of shared SNPs clearly highlight the origin and evolution of the old chalk progenitor and the basalt new sympatric derived species, and their differential environmental stresses (Fig S1A and B), matching the fact that only 28% plants species were present in both the divergent soil types (Hadid et al, 2013). A total of 0.96% and 0.91% SNPs were found in exonic regions of basalt and chalk populations, respectively. About 95.3% from basalt and 95.6% from chalk population of the SNPs are distributed in intergenic regions and 3.74% and 3.49% SNPs are located in regulation regions in basalt and chalk population, respectively (Table S3).

### Genetic divergence of the chalk and abutting basalt populations

A phylogenetic network tree was constructed based on pairwise genetic distance to check the relationship of all the individuals from the chalk and abutting basalt soil populations. We found that all basalt individuals clustered in one clade (marked in red), well separated from the chalk close (marked in blue) in the neighbor-joining tree (Fig 1C). Principal component analysis (PCA) was performed based on the SNP dataset across the whole genome to estimate lineage divergence (Fig 1D). Notably, the chalk and abutting basalt soil populations were fully separated by the first principal component, which explained 26.17% of the variance, whereas the second component explained 19.74% of the variance, which is separated according to soil populations. The PCA, like the phylogenetic tree, displays clear-cut clusters, consistent with their soil origins from divergent rocks and soils (Fig S1A).

Population structure analysis was performed on the two soil wild barley populations to separate the individuals into different numbers of clusters. When the individuals were separated into two groups (K = 2), the individuals from chalk were clearly separated from those from the basalt ones (Fig 1E). Partitioning the individuals into three clusters (K = 3), both of the two soil populations were divided into two subsets, displaying population substructure of each species. These results are consistent with the phylogenetic tree and PCA. All these results were consistent with their divergent ecological origin of chalk and basalt (Fig 1A–E).

Another neighbor-joining tree was constructed based on SNPs from *coding* areas and *noncoding* genomic areas. Individuals from chalk were clustered genetically separately from those of basalt soil (Fig S2), showing the same pattern to that with SNPs across the whole genome (Fig 1C). Notably, the neighbor-joining tree based on *SNPs from noncoding regions mirrors the one from coding regions*, indicating they are under the same selection pressures, past and present. Remarkably, a clear-cut separation is evidenced between the chalk and basalt soil populations. Mutations, including both SNPs and indels from chalk and abutting basalt soil populations were illustrated by circus figure (Fig S3).

### Population genetic diversity and differentiation

Tajima's D highlights that the tested genome regions of chalk and basalt WB populations evolved selectively, nonrandomly, and rejecting neutrality. The peak of D value of the chalk WB population is lower than that of basalt WB population (Fig 2A), suggesting that many basalt population-specific loci (non-overlapping regions on the right in Fig 2A) are under balancing selection, while chalk population-specific loci (non-overlapping regions on the left in Fig 2A) are probably under directional selection. Balancing selection could maintain advantageous polymorphisms and increase genetic diversity (Charlesworth, 2006), that is why the basalt population shows larger genetic diversity, although it is younger than the abutting chalk population. The non-overlapping Tajima's D distribution of the WB populations is similar to that of spiny mice and *Drosophila melanogaster* from EC I at Mount Carmel, where the balancing selection maintains population genetic diversity (Li et al, 2016a).

The decay of linkage disequilibrium (LD) with physical distance of the wild barley populations was estimated by $r^2$. LD of both chalk and basalt populations drop to the bottom within 100 kb (Fig 2B), which is similar to other wild plants, like wild soybean within 27 kb (Zhou et al, 2015), wild rice about 20 kb (Huang et al, 2012), and wild maize about 20 kb (Hufford et al, 2012). As the WB is a highly self-pollinating plant, the LD is on a high level (Flint-Garcia et al, 2003). The genetic diversity of the chalk and abutting basalt wild barley populations measured by Watterson's estimator $\theta_\pi$ was $0.29 \times 10^{-3}$ and $1.13 \times 10^{-3}$, respectively (Fig 2C).

### Genomic divergence

Genomic divergence between chalk and basalt wild barley populations was measured by fixation index, $F_{ST}$ (Fig 2D). As wild barley is mainly self-pollinated, the LD of the basalt population is much

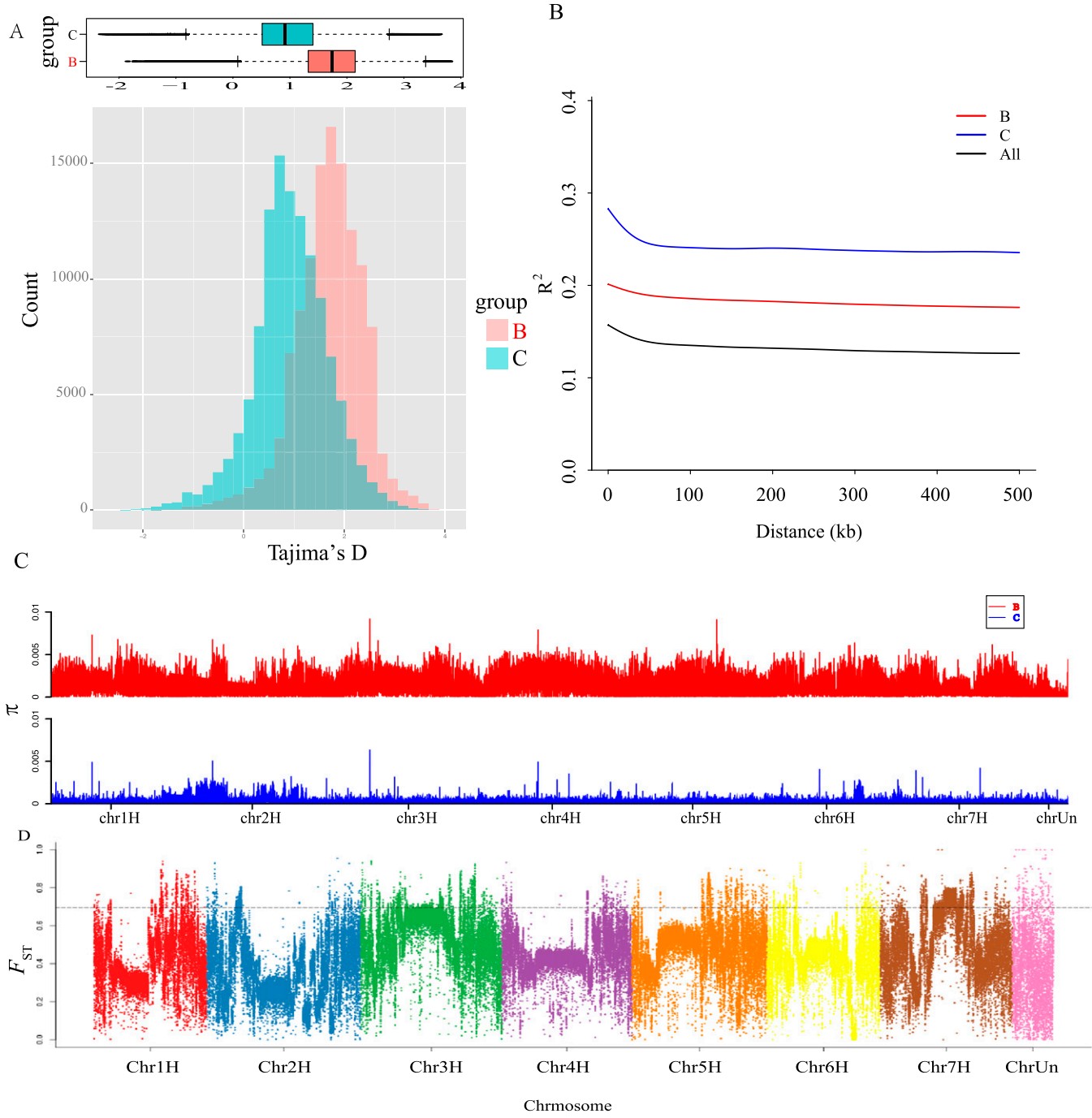

**Figure 2. Wild barley populations diversity and differentiation.**
**(A)** Distribution of Tajima's D from both chalk (in blue) and abutting basalt (in red) wild barley populations. **(B)** Linkage disequilibria of chalk (in blue) and abutting basalt (in red) wild barley populations. **(C)**. Nucleotide diversity ($\theta\pi$) of chalk and abutting basalt populations. **(B, C)** denotes $\pi$ of basalt population in red and (C) is chalk population in blue. **(D)** Genomic differentiation, measured by $F_{ST}$, across each chromosome, y-axis denotes $F_{ST}$ and Chr1~7H and ChrUn on x-axis denotes seven chromosomes and the unknown areas.

lower than that of the chalk population, clearly indicating that the chalk population is under higher linkage disequilibria selection (Fig 2B). Lower LD in basalt population suggests higher recombination rates, which is consistent with higher genetic diversity (Fig 2C) (Lercher & Hurst, 2002). Large divergent loci with $F_{ST} > 0.7$ are

distributed on different chromosomes across the whole genome (Fig 2C). Notably, the lower gene flow and high differential selection between the contrasting and sharply divergent ecological stresses are the main drivers causing the large divergence between the chalk and basalt populations.

### Population demography

The fluctuations of effective populations size of the ancestral population of the current chalk and basalt wild barley populations was examined based on the WGS SNPs. The generation time for wild barley is 1 yr, and the mutation rate is $6.5 \times 10^{-9}$ per site per generation. Both chalk and basalt wild barley populations exhibit the same declining trend from about 200,000 yr ago till very recently, about 4,000 yr ago (Fig 3A), which may be explained by the low temperature during the inter-pluvial cold and dry period (Deuser et al, 1976). There is a small population expansion till now (Fig 3A), possibly because of the increasing temperatures in the Holocene. Both chalk and its derivative abutting basalt populations show the same trend, and the two populations show more or less the same effective population size, probably because they were under the same *macro*-climate. The effective population size of basalt is larger than that of the chalk population just because the volcanic basalt is humid and rich in vegetation (Figs 1A and S1A).

Four demographic models (Fig 4A) were tested. The two populations start to diverge at about $9.1 \times 10^4$ yr ago (Fig 4B), which is parallel to our previous study on blind mole rats, residing in the chalk and basalt at EP (Hadid et al, 2013; Li et al, 2015, 2016b). A new blind mole rat species originated on the Pleistocene basalt, paralleling with the wild barley, studied here (Hadid et al, 2013; Li et al, 2015, 2016b; Šklíba et al, 2016; Lövy et al, 2017). Subterranean blind mole rat, *S. galili*, originated on the Pleistocene volcanic basalt, diverging about $2.28 \times 10^5$ yr ago (Li et al, 2015). The Senonian chalk is older (upper Cretaceous) and has been uplifted with all Israeli mountains in the Neogene, as part of the Alpine mountain formation (Orogenesis).

The best fitting model, # 4, marked by bold font, clearly indicates *primary* isolation in situ on the basalt, accompanied by decreasing

gene flow, rejecting a secondary occurrence from allopatry. This is also clear from the fact that both basalt regions in east Upper Galilee, Alma and Dalton basalt plateaus (see geological map in Fig 1A), are isolated from the Golan basalt plateau, east of the Jordan-Hula rift valley.

Population divergence has been shown in Figs 1C–E and 2D, but the important analysis whether SS diverged *primarily*, that is in situ, or secondarily, that is, occurred *allopatrically*, ex situ first, then established a *secondary* contact remained open. To answer this *critical question*, we conducted fastsimcoal2, *a population demography analysis*, and tested four probable models (Table 1 and Fig 3A). A Allele frequency spectrum approach was used to estimate population demography with fastsimcoal2 (Excoffier et al, 2013). Model 1 was the scenario of allopatric speciation, in which there is no gene flow from the beginning till now; Model 2 stands for splitting of two linages at the beginning because of physical isolation, in other words, without any gene flow, and later secondary contact happened and gene flow occurred; Model 3 was opposite to model 2, in which gene flow happened at the beginning and then stopped. We evaluated model fit based on deltaLikelihood and Akaike information criterion (AIC), and the smallest one was the best fit model. Model 4, speciation with decreasing gene flow, showed the best model fit to our data (Table 1). In primary SS, because there are no physical barriers, gene flow exists at the beginning and decreased because of genetic barrier between the two populations, which is the case of the present study, suggesting *primary* in situ *SS*. The likelihood and AIC help us exclude the possibility of allopatric speciation of Model 1. Gene flow and divergence time were estimated and are shown in Fig 4B. The gene flow from chalk to abutting basalt population initially was $4.12 \times 10^{-4}$, higher than the reverse from the basalt to the chalk population, which was $2.73 \times 10^{-4}$. And later about 5,200 yr ago, the gene flow decreased to $7.63 \times 10^{-5}$ and $1.49 \times 10^{-4}$, respectively (Fig 4B).

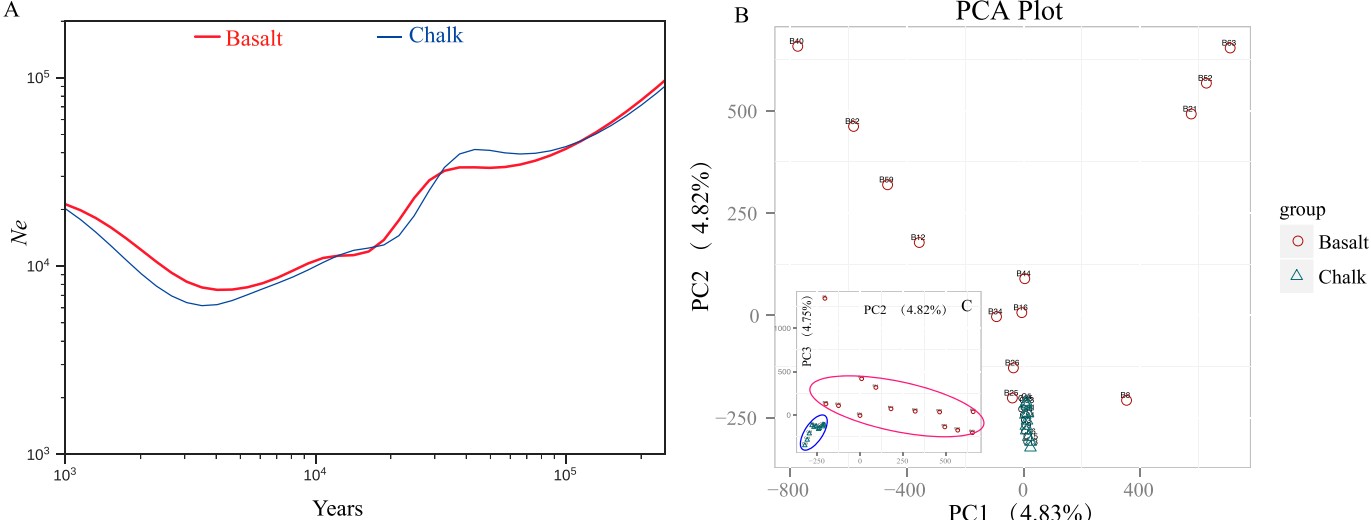

**Figure 3.  Population demography of wild barley populations.**
**(A)** Fluctuation of effective population size of wild barley from chalk and its derivative abutting basalt soil population inferred from SMC++ model. Individuals from basalt population were marked in red and those from chalk were marked in blue, x-axis denotes time, and y-axis denotes effective population size. The two populations show the same declining trend from about 200,000 yr ago, till about 4,000 yr ago. **(B)** Population differentiation of copy number variation from chalk and abutting basalt wild barley populations separated by the first and second principal component. **(C)** Principal component analysis of copy number variation of chalk and basalt wild barley populations, totally separate into two clusters by PC2 and PC3.

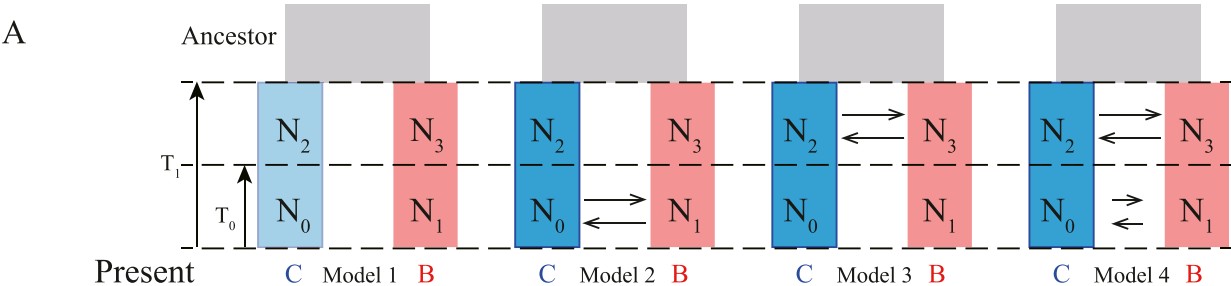

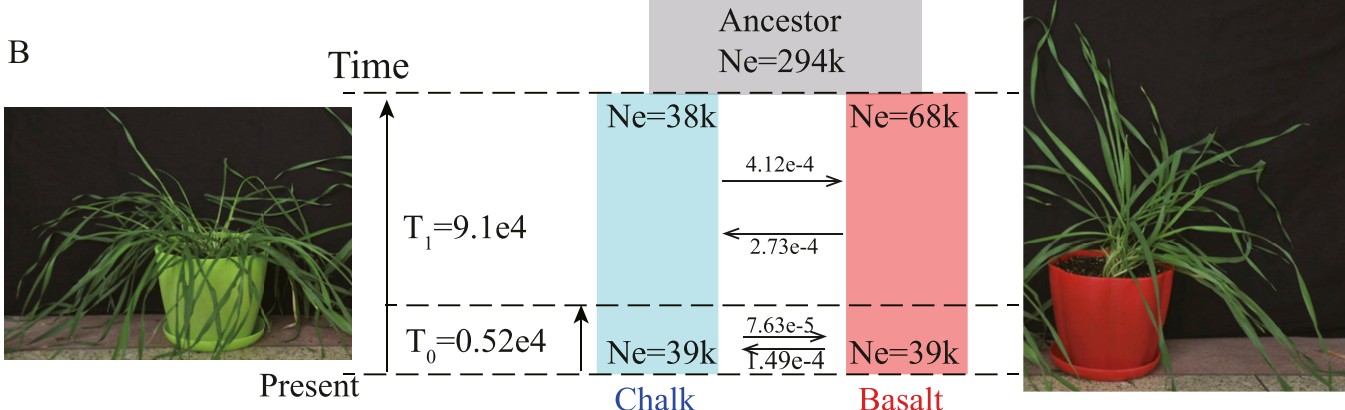

**Figure 4. Coalescence analysis of population demography in wild barley from chalk and derivative basalt.**
**(A)** Four tested probable divergence models. Model 1: speciation without gene flow; Model 2: speciation with recent gene flow following secondary contact; Model 3: speciation with initial gene flow and Model 4: speciation with two distinct periods of higher gene flow at the beginning and lower gene flow recently. **(B)** Parameter estimates for the best fit model of decreasing migration between the chalk and derivative basalt dwelling populations. Notably, the two incipient sympatric species showed phenotypic differences as was shown above. Ne stands for effective population size, and divergence time was marked on the left side by an arrow.

## Natural selection on the two wild barley soil populations

To identify putative selective signals on the two wild barley soil populations during the long period of edaphic adaptation, the genomic regions with large divergence of allele frequency ($F_{ST}$) and high $\log_2 \pi$-ratio of the two populations was estimated (Axelsson et al, 2013). The outliers with $F_{ST} > 0.5$ and $\log_2 (\pi \text{ ratio}) > 0.59$ were considered as putatively selected genes in the chalk population and $F_{ST} > 0.5$ and $\log_2 (\pi \text{ ratio}) > 0.59$ as putative selected genes in the basalt population (marked in red in Fig S4). There are 77 and 184 genes that were selected in chalk and basalt populations, respectively. The distribution of $F_{ST}$ and $\log_2 (\pi_{\text{basalt}}/\pi_{\text{chalk}})$ was shown

in Fig S4. Two genes, *GA20ox-B4* and *GA3ox-D2*, related to GA pathway, were selected in chalk population, and were found to be involved in the production of gibberellins that control many aspects of plant growth and development, including seed germination, stem elongation, leaf expansion, flowering time, and seed development (Sakamoto et al, 2004; Yamaguchi, 2008). As chalk soil is more barren, the growth there should be much affected, and *HORVU3Hr1G064890* gene was selected in the basalt population and which showed resistance to fungi. Basalt is more humid than the dry chalk; hence, both fungi and bacteria are more frequent on basalt as is the case in the EC model (Yin et al, 2015). Another putatively selected gene, CKB4, in basalt population plays an

**Table 1. Summary of model statistics from fastsimcoal2 analyses.**

| Model | No. of parameters | MaxEstLhood | MaxObsLhood | deltaLikelihood | AIC | delta_I(AIC) | Weigthts |
|-------|-------------------|-------------|-------------|-----------------|-----|--------------|----------|
| Model 1 | 7 | −7919541.939 | −1599115.725 | 6320426.214 | 36470852.42 | 9666559.84 | 0 |
| Model 2 | 9 | −5862058.901 | −1599115.725 | 4262943.176 | 26995796.88 | 191504.2956 | 0 |
| Model 3 | 9 | −7767297.873 | −1599115.725 | 6168182.148 | 35769746.59 | 8965454.007 | 0 |
| **Model 4** | **11** | **−5820473.403** | **−1599115.725** | **4221357.678** | **26804292.58** | **0** | **1** |

No. of parameters: how many parameters defined in the simulation; MaxEstLhood: the maximum value for the likelihood if there was a best fit of the expected to the observed SFS.
MaxObsLhood: the maximum likelihood estimated according to the model parameters.
deltaLikelihood: the difference between MaxObsLhood and MaxEstLhood.
AIC, Akaike information criterion; delta_I(AIC), The difference between the smallest and itself.

important role in promotion of photomorphogenesis, suggesting larger biomass (Bu et al, 2011), probably because the basalt soil has higher level of water and nutrition, promoting the WB basalt to robust growth. We have a parallel situation at Evolution Canyon I in Mount Carmel, Israel. This Evolution Canyon I involves two opposite slopes, the "African" hot and dry savannoid, South Facing Slope, versus the "European" cool and humid forested, North Facing Slope, separated by ~250 m. The "European," humid slope, evolved disease resistance in both wild emmer wheat and wild barley against the powdery mildew and leaf rust pathogens. By contrast, the opposite "African slope" evolved  susceptibility in both cereals to the same pathogens (Yin et al, 2015).

### Copy number variation (CNV) divergence between chalk and basalt wild barley populations

The CNVs of the two abutting wild barley soil populations were identified using CNVnator. The number and length of CNVs are larger in basalt than those in chalk (Fig S5A). There are 3,523 CNVs in chalk population with 87.9% of them population specific, and only 1,253 in basalt population with 66.1% of them basalt specific (Fig S5B). The deletions are more frequent than that of duplication in both chalk and basalt populations (Fig S5B). The CNVs are distributed mainly in intergenic regions, but also in exonic and regulation regions (Fig S6). The CNV differentiation between chalk and basalt populations was analyzed using both PCA (Figs 3B and C and S7A) and hierarchical analysis (Fig S7B). In PCA, the two populations were divided by the second principal component (4.82%), and the basalt population possesses larger variation than that of the chalk population, and this contributes to genome variability (Cheng et al, 2005). CNVs from the chalk and basalt populations were separated by both the second and the third principal component (Figs 3C and S7A) and shows two clear-cut clusters. Generally, CNVs from chalk and basalt populations were separated, which could be found also from the hierarchical tree and in SNPs. The presence or absence of CNVs is nonrandom (Egan et al, 2007) as shown in our present data, suggesting the contribution of CNV from each population to adaptation to the local soil environment. Importantly, they could adapt organisms to new environments (Pezer et al, 2015), without a phenotypic effect (Zarrei et al, 2015). Hierarchical tree (Fig S7B) is another way to demonstrate the distribution pattern of CNV between the two soil populations. Generally, chalk and basalt wild barley populations were separated genetically, which was consistent with the ecological separation of other organisms like subterranean mole rats (Hadid et al, 2013; Li et al, 2015, 2016b). The separation of CNV is positively correlated with the divergent selection and adaptation to their environment. Finally, and very important, the patterns of selection of the *coding* and *noncoding* genomes are parallel (Fig S2). Natural selection affects both genomic regions similarly, reinforcing the hypothesis that the *noncoding* genomic regions involve at least partly in regulation like long noncoding RNA (Liu et al, 2015) and are subjected to the natural selection affecting genes coding for proteins.

## Discussion

The evidence presented here clearly highlights that wild barley, the progenitor of cultivated barley, speciates sympatrically on the

Pleistocene basalt, as did the two species *C. hyemalis* in chalk and *C. aleppicus* in basalt (Hadid et al, 2013), and the blind subterranean mole rat *S. galili* (Hadid et al, 2013; Li et al, 2015, 2016b; Lövy et al, 2015). In basalt, our metagenomic bacterial study (in preparation) suggests that EP is a hot spot of SS, from bacteria through plants to mammals, as in Evolution Canyon I (ECI), although the ecological causes are different: *edaphic divergence* in EP and *interslope microclimates* in ECI.

### Genomic comparisons

Following is a brief overview of the current study of WB at EP, indicating the consistency reached by all genomic analyses in the current study, supporting SS of WB at the hot spot of EP. With regard to flowering plants, the species number in the basalt and chalk abutting habitats is 76 and 69, respectively, sharing only 32 plant species (28%). Joint phylogenetic tree of the entire WB genome (Fig 1C) was mirrored by that of the phylogenetic tree of *coding* and *noncoding* genomes (Fig S2), suggesting similar past and present ecological selective pressures causing sharp divergent WB population structure on the abutting divergent ecologies. Remarkably, the pattern of shared and population-specific SNPs of the two soil populations (Fig S1B) also appeared in the SS of subterranean mole rat, *S. galili* at the same locality, dramatically highlighting *interpopulation* SNP separation (Li et al, 2015, 2016b). WB soil populations (Fig 1B) were also clearly separated by PCA analysis (Fig 1D) and structure analysis with K = 2 (Fig 1E). Clearly, the abutting but ecologically divergent chalk-basalt habitats at EP display a hot spot of SS, as was shown in other plants like *C. hyemalis* in chalk and *C. aleppicus* in basalt (Hadid et al, 2013). We also showed earlier by amplified fragment length polymorphism genomic markers that subterranean blind mole rats of the *S. galili* complex (2n = 52) display mammalian microevolution in action on divergent soil types (Polyakov et al, 2004). These involve terra rossa soil on hard Middle Eocene limestone at Rehaniya; rendzina soil on Senonian chalk at Kerem Ben Zimra, and basalt soil on volcanic basalt at Alma and Dalton basalt plateaus. Later, we have shown incipient SS in *S. galili* complex by mitochondrial DNA, mtDNA (Hadid et al, 2013), wide-genome analysis (Li et al, 2015), and transcriptome analysis (Li et al, 2016b), accompanied by ecological behavioral analysis (Lövy et al, 2015, 2017; Šklíba et al, 2016). Although there is ongoing gene flow, because ants and other animals are common and moving seeds in both directions across the 200 m transect, the two WB soil populations diverged into two clear-cut clusters, which occur when the edaphic selection overrides the genetic homogenization of the ongoing gene flow (Nevo, 2011). The genetic diversity in basalt population is higher than that in chalk population nearly across the whole genome (Fig 2C). This is probably because the basalt WB effective population size is larger than that of chalk (Fig 3A), and or, because of higher environmental stresses of hypoxia and hypercapnia during winter flooding on the badly draining basalt because of its clay texture (Singer, 1966). Although the WB population from the higher pH calcareous chalk undergoes heat and drought stresses, that from lower pH siliceous basalt suffered hypoxia and hypercapnia (Brown et al, 1978; Nevo, 1999; Li et al, 2016b). Both are stressful, but the WB growing in winter may suffer more from hypoxia and hypercapnia, hence the high genetic diversity, possibly

together with population size. Notably, the basalt soil is humid, cool, and comparatively rich in nutrition, which will facilitate population expansion. These differences, especially in temperature, hotter chalk, and cooler basalt, might explain the earlier flowering of WB on the chalk population and later flowering of the basalt population leading to *premating reproductive isolation* as was shown in *Ricotia lunaria* at EC because of differential temperatures of the African and temperate ecotypes when grown in a common garden experiment (Qian et al, 2018). Similarly, wild barley on dry terra rossa matures much earlier than on the abutting humid basalt at Evolution Slope (Tabigha) (Wang et al, 2018). Likewise, the differential levels of gibberellins in chalk–basalt populations of WB affect their differential flowering time, that is, increase reproductive isolation.

The LD of the chalk WB population is much higher than that of the basalt population, clearly indicating that the chalk population is under higher linkage disequilibria selection (Fig 2A). Lower LD in basalt population suggests higher recombination rates, which is consistent with its higher genetic diversity (Fig 2C) (Lercher & Hurst, 2002). The LD of the basalt population is much lower than that of the chalk population, clearly indicating that the chalk population is under higher linkage disequilibria selection (Fig 2A). Lower LD in basalt population suggests higher recombination rates, which is consistent with higher genetic diversity (Fig 2C) (Lercher & Hurst, 2002). Large genetic divergence $F_{ST}$ between chalk and basalt WB populations across chromosomes was probably due to the sharp ecological divergence (Fig 2D). Tajima's D (Tajima, 1989) highlights that the tested genome regions of chalk and basalt WB populations evolved *selectively*, *nonrandomly*, and *rejecting neutrality*. The peak of D value is lower in chalk than in basalt WB (Fig 2A), suggesting that many basalt WB-specific loci are under balancing selection, whereas chalk population–specific loci are probably under directional selection. Balancing selection could maintain advantageous polymorphisms and increase genetic diversity (Charlesworth, 2006), that is why the basalt population shows larger genetic diversity, although it is younger than the abutting chalk population. The non-overlapping Tajima's D distribution of the WB populations is similar to that of spiny mice and *D. melanogaster* from EC I at Mount Carmel, where the balancing selection maintains population genetic diversity (Li et al, 2016).

## Population demography

Both chalk and basalt WB populations exhibit the same declining trend from about 200,000 yr till about 4,000 yr ago (Fig 3A), probably caused by the low temperature during the inter-pluvial cold and dry period (Deuser et al, 1976). A small population expansion of both populations till now (Fig 3A) was probably caused due to increasing temperatures in the Holocene. The two populations showing similar effective population size, *N*e, due to similar *macro*-climate. The two populations start to diverge at about $9.1 \times 10^4$ yr ago (Fig 4B), which is parallel to our previous study on blind mole rats, residing in the chalk and sympatrically speciating on basalt at EP, originating $2.28 \times 10^5$ ya (Hadid et al, 2013; Li et al, 2015, 2016b; Šklíba et al, 2016; Lövy et al, 2017). The two basalt plateaus of Alma and Dalton (see geological map in Fig 1A) are Pleistocene in age and are about 1.6 ~ 1.8 million yr ago (Mor & Steinitz, 1984). This volcanic eruption, opened the new

basalt niche domain, in which fungi (Grishkan et al, 2008), animals such as blind mole rat *S. galili* (Hadid et al, 2013; Li et al, 2015, 2016b, Lövy et al, 2015, 2017) and plants, such as the wild barley discussed here and other plants, such as *C. hyemalis* in chalk and *C. aleppicus* in basalt (Hadid et al, 2013) makes EP a hot spot of SS. The Pleistocene basalt at EP, initiated a wave of SS across life. We have currently showed it also in bacteria (in preparation).

SS has always been contentious (Jiggins, 2006; Bolnick & Fitzpatrick, 2007), and in the genome era now, it is still controversy in whether it is primary speciation or secondary contact (Richards et al, 2019). As the basalt soil in upper Galilee erupted from a volcano, it seems like a basalt "island" on chalk soil "ocean" (Geological map in Fig 1A). Therefore, the basalt population is later and derivative of the chalk population, and shares the common ancestor. The demographic analysis of population divergence implemented by fastsimcoal2 provides a strong support to the hypothesis that the chalk and abutting derivative basalt populations split primarily combined with decreasing gene flow (Fig 4B). These results suggest that the two populations unfold primary SS with gene flow, and reject the secondary contact hypothesis from allopatry (Fig 4A, model 1). This is also the case in EC I in the dramatic SS of three species of wild emmer wheat the progenitor of bread wheat (Wang et al, 2020), and other species that speciated sympatrically from bacteria to mammals (Nevo, 2014). The best fitting model, suggesting *primary* in situ SS, gene flow, and divergence time, were estimated and shown in Fig 4B.

## CNV differs in chalk and basalt WB

CNVs from the chalk and basalt show two clear-cut clusters (Figs 3C and S7A). Generally, chalk and basalt wild barley populations were separated genetically, which was consistent with the ecological separation at EP of other organism (Hadid et al, 2013; Li et al, 2015, 2016b). Notably, the separation of CNV is positively correlated with the divergent selection and adaptation to their contrasting chalk-basalt ecologies (Fig 1C).

## Putatively selected genes and reproductive isolation in SS of WB

We identified putatively selected genes, 77 and 184 genes in chalk and basalt populations, respectively (Fig S4). Two genes, *GA20ox-B4* and *GA3ox-D2*, related to GA pathway, were selected in chalk population, involved in the production of gibberellins that control many aspects of plant growth and development, including seed germination, stem elongation, leaf expansion, flowering time, and seed development (Sakamoto et al, 2004; Yamaguchi, 2008). As chalk soil, rendzina, is more barren, the growth there should be much affected. *HORVU3Hr1G064890* gene was selected in basalt population and show resistance to fungi. Basalt is more humid than the dry chalk, hence both fungi and bacteria are more frequent on basalt as is the case in the EC model (Yin et al, 2015). Another putatively selected gene, CKB4, in basalt population, plays an important role in promotion of photomorphogenesis, suggesting larger biomass (Bu et al, 2011), probably because the basalt soil has higher level of water and nutrition, promoting robust WB growth, as occurs in wild emmer wheat and wild barley from the humid European, NFS at EC I, Mount Carmel, evolving *resistance* to powdery mildew and leaf rust fungi, and *susceptibility* to these fungi on the opposite dry abutting south-facing AS.

### Is SS *rare* or *common*?

The discussion about the abundance of SS as a model for the origin of new species first suggested in principle by Darwin, is ongoing despite mounting of theoretical and empirical evidence supporting it (Appendix data). We have been studying the SS model since 1990 in two basic models. First, the EC model with four replicates in the mountains of Carmel, Galilee, Negev Desert, and Golan (EC I–IV) (Fig S8). This model, best represented by EC I and EC II, is a *microclimatic* model where two abutting slopes, tropical, hot, dry, and savannoid south-facing slope (SFS), also dubbed the AS, abuts with a temperate, cool, and humid forested NFS, also dubbed the European Slope (ES), separated a few meters to hundreds of meters from the SFS = AS (Fig S9). The SFS-AS receives at EC I 200–800% more irradiance from the sun shining on it all year round, leading to high solar radiation, temperature and drought (Pavlícek et al, 2003), hence the savannoid African biome slope. By contrast, the NFS = ES is shaded from direct irradiance and characterized by 7% more humidity than the SFS = AS, hence forested. In 2003, we embarked upon a microsite divergent edaphically, dubbed "EP," deeply studied for SS in the blind mole rat, *S. galili* (Hadid et al, 2013; Li et al, 2015, 2016b, Lövy et al, 2015, 2017), and the current study on wild barley, *H. spontaneum*, the progenitor of cultivated barley, which originated a new species of WB on the basalt, adapted to a siliceous world of volcanic basalt, derived from the calcareous chalk progenitor. In a parallel article, we described how wild barley initiated formation of a new species of wild barley at EC I, together with other six species from bacteria to mammals (Nevo, 2014; Qian et al, 2018; Wang et al, 2020).

These two microsites, with sharply *divergent ecologies*, *microclimatic* at "EC I," and *edaphic* in "EP," are *hot spots* of *primary incipient or full* SS across life, from bacteria to mammals, as already demonstrated in EC I (Nevo, 2014). This is also true at EP, as our current ongoing bacterial project suggests. We conclude that SS, far from being a *rare* model of species origin, is a *common* model, since "ECs" in sharply ecologically divergent microsites, abound on our planet caused by geologic, edaphic, climatic, biotic, and abiotic contrasts. Such microsites represent *evolution in action* across life, where both adaptive evolution and incipient or full SS abound, *actively* generating continuously new species evolving from *primary* SS as shown in the present study at EP, and dramatically in the evolution of three species of wild emmer wheat at EC I (Wang et al, 2020). The vision of Darwin in 1859 has been substantiated in microsites where sharp ecological divergence in free breeding populations with gene flow (Brown et al, 1978) and two contrasting ecologies occur. They clearly indicate how selection overrules the homogenizing effects of gene flow across life from viruses (unpublished) and bacteria to mammals (Nevo, 2011).

# Materials and Methods

### Sample collection and sequencing

The present study was conducted in EP, eastern Upper Galilee, Israel (Fig 1A, the geological map [coordinates: 33.0463° N, 35.4929° E]), where the original soil is rendzina weathered on Senonian chalk covered by a Pleistocene volcanic basalt eruption, forming the Alma and Dalton basalt plateaus, weathered to basalt soil. Of 113 plant species totally, the abutting chalk rock (rendzina soil) and basalt rock (basalt soil) share only 32 plant species (28%). Remarkably, 44 plants species (39%) and 37 plant species (33%) were unique to the basalt and chalk, respectively (Fig 1A). The wild barley *H. spontaneum* (Fig 1B) was collected from chalk and abutting basalt soils, respectively, in 2016. We analyzed genomically 13 wild barley genotypes from basalt and 11 genotypes from the abutting chalk.

### Library preparation

A total amount of 1.5 μg genomic DNA per sample was used as input material for DNA sample preparation. Sequencing libraries were generated using Truseq Nano DNA HT sample preparation kit (Illumina) following the manufacturer's instructions, and index codes were added to attribute sequences to each sample. Briefly, the DNA sample was fragmented by sonication to a size of around 350 bp, and then DNA fragments were end polished, A-tailed, and ligated with the full-length adapter for Illumina sequencing with further PCR amplification. At last, PCR products were purified (AMPure XP system), and libraries were analyzed for size distribution by Agilent2100 Bioanalyzer and quantified using real-time PCR.

### Genome sequencing and quality control

Pair-end libraries were constructed with 150-bp insert and sequenced on Illumina HiSeq X10 for all samples. After germination in the laboratory, leaves of each individual were harvested. A total of 23 individuals, including 12 from basalt and 11 from chalk were collected. DNA was extracted from basalt and chalk population using (cetyl trimethylammonium bromide) method, respectively. DNA libraries with 350-bp insert were constructed using kit from Illumina and pair end sequencing of 150 bp were applied on HiSeq ×10. The clean reads, generated after filtration of adapters and low quality reads, were mapped against the reference genome (Mascher et al, 2017). Whole genome resequencing of wild barley from two genealogical populations were performed on the Illumina HiSeq2500 platform. In total, 5.53 Tb raw data were generated for two edaphic populations. First, low-quality pair-end reads (reads with ≥10% unidentified nucleotides [N]; >10 nt aligned to the adapter, allowing ≤10% mismatches; >50% bases having Phred quality <5) and putative PCR duplicates generated in the library construction process were removed from downstream analyses.

### Reads mapping and SNP calling

The remaining high-quality reads paired-end reads were mapped to the reference genome *Hordeum vulgare* (Mascher et al, 2017) using Burrows-Wheeler Aligner (Li & Durbin, 2009) (Version: 0.7.17) with the command "mem -t 4 -k 32 −M." To reduce mismatch generated by PCR amplification before sequencing, duplicates were removed from downstream analyses using Picard (https://broadinstitute.github.io/picard/).

SNP calling was performed after reads alignment on a population scale using a Bayesian approach as implemented in the package SAMtools (Li et al, 2009). We then calculated genotype

**Life Science Alliance**

likelihoods from reads of each individual, and the allele frequencies in the sample with a Bayesian approach. The "mpileup" command was used to identify SNPs with the parameters as "-q 1 -C 50 -S -D -m 2 -F 0.002 –u." Then, to exclude SNP calling errors caused by incorrect mapping or indels, only high-quality SNPs (coverage depth ≥5 and ≤100, RMS mapping quality ≥20, maf ≥ 0.05, miss ≤ 0.1) were kept for subsequent analysis.

### Phylogenetic tree and population structure

To clarify the phylogenetic relationship from a genome-wide perspective, an individual-based neighbor-joining tree was constructed based on the p-distance using the software TreeBest v1.9.2 (http://treesoft.sourceforge.net/treebest.shtml). The population genetic structure was examined via an expectation maximization algorithm, as implemented in the program FRAPPEv1.170 (http://med.stanford.edu/tanglab/software/frappe.html). The number of assumed genetic clusters K ranged from two to seven, with 10,000 iterations for each run. We also conducted PCA to evaluate genetic structure using the software genome-wide complex trait analysis (GCTA) (Yang et al, 2011).

### Genetic diversity and LD

Nucleotide diversity ($\theta\pi$) and Watterson's estimator ($\theta$w) were calculated based on the high-quality SNPs of each soil population using the sliding-window approach (100-kb windows and step size of 40 kb). To estimate and compare the pattern of LD for each of the two soil populations, the squared correlation coefficient ($r^2$) between pairwise SNPs was computed using the software Haploview v4.269. Parameters in the program were set as: "-n -dprime -minMAF 0.05." The average $r^2$ value was calculated for pairwise markers in a 20-kb window and averaged across the whole genome. We found differences in the rate of decay and level of LD value, which reflected variations in population demographic history and effective population size ($N_e$) between the two populations.

### Identification of putatively selected genes

To identify genome-wide selective sweeps associated with edaphic adaptation, we calculated the Tajima's D and the genome-wide fixation index ($F_{ST}$) for the population pairs using a sliding-window method with 100-kb window size and 40-kb increments. The genome-wide distribution of $F_{ST}$ value was calculated by vcftools (Danecek et al, 2011). We considered the windows with the top 5% $F_{ST}$ and 5% largest and smallest Tajima's D simultaneously as candidate outliers under strong selective sweeps. All outlier windows were assigned to corresponding SNPs and genes.

### Effective population size fluctuation and population demography

The fluctuation of effective population size of the chalk and its derivative basalt dwelling wild barley were estimated using SMC++ (Terhorst et al, 2017), which was based on sequential Markov coalescent method. This method is based on whole genome SNPs which could avoid errors induced by phasing (Terhorst et al, 2017). Two replicated selections were used in each group and the non-effective regions (smaller than five or larger than sequencing depths) were masked with parameter "vcf2smc –m." Every individual was set as one of the distinguished pair to generate varied independently evolving sequence, and all the chromosome samples were used as input when running SMC++ estimate. A mutation rate of $3.5 \times 10^{-9}$ was used for demographic estimation, and each generation last for 1 yr.

To estimate the demography of the chalk and derivative basalt soil populations, we constructed folded site frequency spectrum (SFS) with neutral sites that from intergenic areas by using the python script easySFS (https://github.com/isaacovercast/easySFS). The 1,306,327 neutral sites from intergenic regions were selected based on the balance of number of segregating sites and missing data. The best model was selected using the model-based composite likelihood method implemented in fastsimcoal 2.6 (Excoffier et al, 2013). And in this model, both the likelihood and AIC were calculated for a simulated SFS under a specified demographic scenario, and then were compared with that of the observed SFS to evaluate the model fit. We tested four most probable demographic models (Fig 4A) to estimate the divergence scenario of the two soil populations. These models including the following: (1) speciation without any gene flow, this should be a totally allopatric speciation; (2) speciation with recent gene flow, in other words, diverged allopatrically at the beginning with secondary contact; (3) early gene flow between the two soil populations but split immediately with some physical barriers; and (4) speciation with two distinct periods of migration, larger gene flow at the beginning and lower gene flow very recently. We performed 100 independent runs for each of the above model to choose the best model and parameter estimation in Fsc. Each run with 100,000 coalescent simulations and 50 expectation maximization cycles for parameter maximization.

Parametric bootstrapping (100 simulated datasets under the best model with 50 rounds of parameter estimation) was carried out to estimate 95% confidence intervals for each parameter for the best fit model.

CNV was identified from each of two wild barley populations by CNVnator (Abyzov et al, 2011). The ratio of average read-depth signal to its SD should be between 4 and 5, which help us decide the optimal bin size and the final bin size for the present study was from 100 to 1,000 bp. The calls on the genome without annotation information were removed from downstream analyses. All the calls less than 1 kb were also excluded. The CNVs from each individual of the same population were merged into unique CNVRs with BEDtools (Quinlan & Hall, 2010).

### Data accession

The sequencing data from this publication have been deposited to the National Center for Biotechnology Information (NCBI) database (https://www.ncbi.nlm.nih.gov/) and assigned the identifier PRJNA622206."

## Supplementary Information

## Acknowledgements

This project was supported by the earmarked fund for China Agriculture Research System (CARS-5), Lanzhou University's "Double First-Class" Guided

Project-Team Building Funding-Research Startup Fee for K Li and the Ancell-Teicher Research Foundation for genetics and molecular evolution to E Nevo.

## Author Contributions

K Li: resources, data curation, software, and formal analysis.
X Ren: data curation and investigation.
X Song: data curation, formal analysis, and investigation.
X Li: data curation and formal analysis.
Y Zhou: software.
E Harlev: resources.
D Sun: conceptualization and funding acquisition.
E Nevo: conceptualization, resources, supervision, and writing—review and editing.

## Conflict of Interest Statement

The authors declare that they have no conflict of interest.

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
