## [Reviewer comments · Life Science Alliance]

Life Science Alliance

Incipient sympatric speciation in wild barley caused by geological-edaphic divergence

Kexin Li, Xifeng Ren, Xiaoying Song, Xiujuan Li, Yu Zhou, Eli Harev, Dongfa Sun, and Eviatar Nevo

DOI: [10.26508/lsa.202000827](https://doi.org/10.26508/lsa.202000827)

Corresponding author(s): Eviatar Nevo, University of Haifa+Institute of Evolution and Dongfa Sun, Wuhan Agriculture University

Review Timeline:

Submission Date:	2020-06-22
Editorial Decision:	2020-07-23
Revision Received:	2020-08-20
Editorial Decision:	2020-08-21
Revision Received:	2020-09-18
Accepted:	2020-09-23

Scientific Editor: Shachi Bhatt

Transaction Report:

July 23, 2020

Re: Life Science Alliance manuscript #LSA-2020-00827-T

Prof. Eviatar Nevo
University of Haifa+Institute of Evolution
199 Aba Khoushy Ave., mount carmel
Haifa, 3498838
Israel

Dear Dr. Nevo,

Thank you for submitting your manuscript entitled "Sympatric speciation in wild barley caused by geological-edaphic divergence" to Life Science Alliance. The manuscript was assessed by expert reviewers, whose comments are appended to this letter.

While the reviewers find your results showing the evolution of a new lineage of wild barley interesting and important, they note a lack of clarity in the experimental details, and a need to adjust the interpretation of the results in keeping with the data provided. These concerns include, but are not limited to, removing the speculation regarding speciation from the title, abstract, and text, and modifying the text to focus on the conclusions that are supported by the analyses. Please also include a proper Methods section in the main text of your revised manuscript, which should be sufficiently detailed so as to render the study reproducible. Finally, please address reviewer 3's minor concerns and the editorial points below.

In our view these revisions should typically be achievable in around 3 months. However, we are aware that many laboratories cannot function fully during the current COVID-19/SARS-CoV-2 pandemic and therefore encourage you to take the time necessary to revise the manuscript to the extent requested above. We will extend our 'scoping protection policy' to the full revision period required. If you do see another paper with related content published elsewhere, nonetheless contact us immediately so that we can discuss the best way to proceed.

Please indicate the meaning of the bold text in the Table 1 legend.

Please modify: \log_2 (2 subscript)

Please note that papers are generally considered through only one revision cycle, so strong support from the referees on the revised version is needed for acceptance.

Thank you for this interesting contribution to Life Science Alliance. We are looking forward to receiving your revised manuscript.

Sincerely,

Reilly Lorenz
Editorial Office Life Science Alliance
Meyerhofstr. 1
69117 Heidelberg, Germany
t +49 6221 8891 414
e contact@life-science-alliance.org
www.life-science-alliance.org

B. MANUSCRIPT ORGANIZATION AND FORMATTING:

Reviewer #1 (Comments to the Authors (Required)):

I think the paper should be published, perhaps using a different title. I see no evidence of speciation, but population isolation and differential selection due to environmental differentiation is indeed obvious. Unfortunately for one of the primary tenets of this paper, it's also quite common.

Reviewer #2 (Comments to the Authors (Required)):

This is an important contribution based upon a very useful environment for study of evolution in wild plant populations. The manuscript lacks a methods section. More details of the experimental materials and methods would aid in understanding the work. Other aspects of the manuscript are all satisfactory.

Reviewer #3 (Comments to the Authors (Required)):

This MS presents significant genomic evidence for the incipient microevolution of a new lineage of Wild Barley (WB) at "Evolution Plateau" (EP). The phylogenetic and statistical analysis clearly shows the separation of WB into two differentially selected populations on the chalk and basalt regions of the EP site. However, the data are not yet sufficient to conclude that a truly novel species has formed. As their own analysis in Table 1 and Figure 4 indicates, there is evidence for continuing but reduced genetic exchange between the two populations. No attempts at hybridizing the two populations are reported. In my opinion, the incomplete nature of the genetic separation process should be incorporated into the title, abstract and text. For example, adding the phrase "Early stages of" at the start of the title would produce a more accurate statement of the conclusions that can legitimately be drawn from the sequence data analysis presented in the MS.

The research presented here depends on a number of generic sequence data analysis tools. It would make the paper clearer for the reader to explain the logic and significance of each analytical procedure when it first appears in the text. Naming the tool and providing a reference are not adequate.

Line 18: add (SS) after "sympatric speciation"

Line 22: "separation" should read "SS separating"

Line 30: "were" for "was"

Line 44: "illuminate" for "highlight"

Line 48: "might it" for "it might"

Line 98: "shared" in what way(s)?

Table S3: define "lv" and "ls"

Line 120: define "PCA" (principle component analysis)

Line 135: What is the parameter k and what do the different colors indicate in Fig 1E histograms?

Fig S3: The chromosome comparisons seem to show more similarity than difference between the two populations. These data require a more detailed discussion and presentation. It would be appropriate to have quantitative figures on the indels similar to those for SNPs. Since these will come later, mention that fact here.

Line 184: The computation of effective population size based on SNP data needs more explanation.

Line 220: The computation and meaning of the fastsimcoal2 numbers in Table 1 requires explanation.

Line 221: #14 should be #4.

Lines 221-233: The logic of this argument needs to be articulated more full, both for determining "best fit" and for the exclusion of allopatry.

Lines 254-255: Fig 3 does not indicate individual genetic loci. Shouldn't the authors instead refer to Figure 2 indicating distinct chromosomal locations and the parameters discussed in the text?

Line 274: delete both "are"s.

Lines 277ff: The statistical analysis really adds no functional or ecological information and can be removed.

Lines 292-296: This is an important observation. It could be supported by citing one or more articles on the regulatory roles played by short and long non-coding RNAs in plants, particularly those affecting developmental timing.

Line 308: "Floristically" is not proper English usage. An equivalent formulation would be "With regard to flowering plants,"

Lines 348-349: Are there direct measurements of flowering time to confirm this inference from the GA-related sequences?

Line 367: Is there genetic data (not geological) indicating the basalt population is younger than the chalk population?

Line 385: Cite Hadid 2013 for Crocus.

Line 405: Is Fig 3B or Fig 2B the correct reference here?

Line 428: "controversy" reads better as "debate"

Line 443: "Originated" a new species of WB may be an overinterpretation of the evidence presented here. "Initiated formation" of a new species would be better.

Line 459: It's questionable that "free breeding" is the right description for self-fertilizing plants such as WB.

Please indicate the meaning of the bold text in the Table 1 legend.

It was described in line 211.

Please modify: \log_2 (2 subscript)

Yes, modified.

Reviewer #1 (Comments to the Authors (Required)):

I think the paper should be published, perhaps using a different title. I see no evidence of speciation, but population isolation and differential selection due to environmental differentiation is indeed obvious. Unfortunately for one of the primary tenets of this paper, it's also quite common.

My colleague, Dr Kexin Li, will resubmit the MS LSA-2020-00827-T with all needed revision. I wish to briefly explain, as also shortly explained in the summary blurb below, why we retain in the title, abstract, and main text, the concept of “**sympatric speciation**” although we added for clarification *incipient sympatric speciation*, across the MS, and fully explained the issue in our discussion.

The concepts of biological species and speciation are still contentious despite the long span since the origin of species have been deeply theorized by Darwin, and since he suggested, in principle, that new species can originate in an interbreeding metapopulation with gene flow (although his phrasing was different and did not use the term sympatric speciation (SS)).

Our Institute of Evolution (IOE) at the University of Haifa Israel, established by me in 1972, and particularly my lab of “Evolutionary Biology”, one of 25 research labs in IOE, and previously myself since 1948, focused on the origin of species across life,

from bacteria to mammals, primarily on the *Spalax ehrenbergi* superspecies, and later across life in the Evolution Canyon model. (See my full list of publications, and particularly the list of “Evolution Canyon” (EC I) at <http://evolution.haifa.ac.il>. On EC we published ~ 250 papers. We initiated the long-term project of the Evolution Canyon model (EC) in microsites divergent *microclimatically* (tropical hot, dry, savannoid versus temperate, cool, humid and forested, abutting at the base of the canyons meters apart, extending in midslopes and top- slopes to hundreds of meters apart (see Fig. 1). We initiated this long-term project in 1990 in four microsites in Israel, in the mountains of Carmel, Galilee, Negev, and Golan (EC I-IV). We extended these microsites to two *edaphically divergent* microsites, dubbed “Evolution-Plateau” and “Evolution Slope”, both in Upper Galilee. We identified in ECI, Mount Carmel, and in “Evolution Slope” hot spots of incipient or full sympatric speciation, based on both prezygotic and postzygotic reproductive isolation, 7 species that underwent SS at ECI, and till now two species, first in subterranean mammals *Spalax galili*, the second in wild barley *Hordeum spontaneum* (the current MS ISA-2020-00827-T) and about 20 species of soil bacteria (in preparation). In ECI, at Mount Carmel, we identified 7 species (out of 2500 species just recorded) that speciated either *incipiently* or fully sympatrically. The list of these species appear in the List below including soil bacterium, *Bacillus simplex* (1), wild barley *Hordeum spontaneum* (2), wild emmer wheat, *Triticum dicoccoides* (3), the crucifer *Ricotia lunaria* (4), beetle, *Oryzaphilus surinamensis* (5), fruitfly *Drosophila melanogaster* (6), and spiny mouse, *Acomys cahirinus* (7). At “Evolution Plateau” we identified a new species of subterranean mammals in the superspecies *Spalax ehrebergi*, specifically in *Spalax galili*, on which we wrote 7 papers listed below after the list of EC I. All these species that we identified as *incipiently* or fully sympatrically speciating, are very different than just **population divergence**. They showed either genetically, genomically, or experimentally, prezygotic and postzygotic reproductive isolation, clearly associating with SS. The chalk and basalt are *abutting*. Like gene flow is still going on between the two populations on chalk

and basalt dramatically divergent in environmental stresses, hypoxia levels and food resources supported by bitter taste receptors.

Summary blurb of MS Lsa-2020-00827-T

Sympatric speciation is still contentious but here based on genome-wide analysis, we show incipient sympatric speciation of an emerging new wild barley species from *Hordeum spontaneum*, the progenitor of all cultivated barleys, at "Evolution Plateau" (EP), eastern Upper Galilee, Israel.

Reviewer #2 (Comments to the Authors (Required)):

This is an important contribution based upon a very useful environment for study of evolution in wild plant populations. The manuscript lacks a methods section. More details of the experimental materials and methods would aid in understanding the work. Other aspects of the manuscript are all satisfactory.

The materials and methods parts were transferred from the supplement to the main text.

Reviewer #3 (Comments to the Authors (Required)):

This MS presents significant genomic evidence for the incipient microevolution of a new lineage of Wild Barley (WB) at "Evolution Plateau" (EP). The phylogenetic and statistical analysis clearly shows the separation of WB into two differentially selected populations on the chalk and basalt regions of the EP site. However, the data are not yet sufficient to conclude that a truly novel species has formed. As their own analysis in Table 1 and Figure 4 indicates, there is evidence for continuing but reduced genetic exchange between the two populations. No attempts at hybridizing the two populations are reported. In my opinion, the incomplete nature of the genetic separation process should be incorporated into the title, abstract and text. For example, adding the phrase "Early stages of" at the start of the title would produce a more accurate statement of the conclusions that can legitimately be drawn from the sequence data analysis presented in the MS.

Please find our responses to reviewer 1

The research presented here depends on a number of genetic sequence data analysis tools. It would make the paper clearer for the reader to explain the logic and significance of each analytical procedure when it first appears in the text. Naming the

tool and providing a reference are not adequate.

Detailed methods were moved from the supplementary to the main text, and readers can find and repeat our data analyses.

Line 18: add (SS) after "sympatric speciation"

Yes, added

Line 22: "separation" should read "SS separating"

Yes, corrected

Line 30: "were" for "was"

Yes, corrected

Line 44: "illuminate" for "highlight"

Yes, corrected

Line 48: "might it" for "it might"

Yes, corrected

Line 98: "shared" in what way(s)?

It was changed to "present in both"

Table S3: define "lv" and "ls"

It was corrected to ts, transition; lv to tv: transversion;

Line 120: define "PCA" (principle component analysis)

This was defined in line 105

Line 135: What is the parameter k and what do the different colors indicate in Fig 1E histograms?

k was the number of clusters and was defined The different colors in Fig 1E were explained in the legend to the Figure

Fig S3: The chromosome comparisons seem to show more similarity than difference between the two populations. These data require a more detailed discussion and presentation. It would be appropriate to have quantitative figures on the indels similar to those for SNPs. Since these will come later, mention that fact here.

Yes, most of the genome are is the same between the chalk and basalt populations of wild barley. compared to the whole genome, only a few loci are mutated between the soil populations. we have a venn figure showing the

number of unique and shared SNPs of the two populations. While fig S3B was removed from the manuscript.

Line 184: The computation of effective population size based on SNP data needs more explanation.

Sure, the mutation rate and generation time were added.

Line 220: The computation and meaning of the fastsimcoal2 numbers in Table 1 requires explanation.

We add more explanations just below the table

Line 221: #14 should be #4.

Yes, corrected

Lines 221-233: The logic of this argument needs to be articulated more full, both for determining "best fit" and for the exclusion of allopatry.

We added more explanation on how to choose the best model and how to exclude the possibility of allopatry.

Lines 254-255: Fig 3 does not indicate individual genetic loci. Shouldn't the authors instead refer to Figure 2 indicating distinct chromosomal locations and the parameters discussed in the text?

Fig. S3 explains the difference of different loci, and the citation was changed to Fig. S3

Line 274: delete both "are"s.

Yes, deleted.

Lines 277ff: The statistical analysis really adds no functional or ecological information and can be removed.

Although we don't know the functional or ecological information of the CNV, this is a description of CNV of the two populations and future work might illuminate it.

Lines 292-296: This is an important observation. It could be supported by citing one or more articles on the regulatory roles played by short and long non-coding RNAs in plants, particularly those affecting developmental timing.

Yes, add citation on long non-coding RNA.

Line 308: "Floristically" is not proper English usage. An equivalent formulation would be "With regard to flowering plants,"

Yes, corrected.

Lines 348-349: Are there direct measurements of flowering time to confirm this inference from the GA-related sequences?

Not yet. They will be sought in future work.

Line 367: Is there genetic data (not geological) indicating the basalt population is younger than the chalk population?

No.

Line 385: Cite Hadid 2013 for Crocus.

Yes, cited.

Line 405: Is Fig 3B or Fig 2B the correct reference here?

Fig. 3B is the correct one as it describes CNV differences between the two species populations.

Line 428: "controversy" reads better as "debate"

Yes, changed to CONTROVERSY

Line 443: "Originated" a new species of WB may be an overinterpretation of the evidence presented here. "Initiated formation" of a new species would be better.

Yes, was changed to initiated formation of a new species

Line 459: It's questionable that "free breeding" is the right description for self-fertilizing plants such as WB.

Although the wild barley is self-fertilizing species, there is still gene flow between populations, see **brown et al 1978 on outcrossing in barley genetic newsletter where we showed that outcrossing in wild barley increases from the Negev desert from 0.5% to a higher level of outcrossing up to 9.6%**

.

August 21, 2020

RE: Life Science Alliance Manuscript #LSA-2020-00827-TR

Prof. Dongfa Sun
College of Plant Science and Technology, Huazhong Agricultural University
1 shizishan Ave. Hongshan,
Wuhan 430072
China

Dear Dr. Sun,

Thank you for submitting your revised manuscript entitled "Incipient sympatric speciation in wild barley caused by geological-edaphic divergence". We would be happy to publish your paper in Life Science Alliance pending final revisions necessary to meet our formatting guidelines.

Along with the changes listed below, please also make the following edits in the revised manuscript,

Callouts -

--Supplemental figure S6A and S6B are called out separately in the manuscript text, but the S6 figure and S6 figure legends do not include an A or B - please reconcile

Author Contributions

-- There is an author Wang Y. that is listed in the 'Author Contributions' but is missing from the Author List, both in eJP and in the manuscript file
-- Cui X. and Li X. are listed on the author list, but not included in the Author contribution. Please clarify what their role was in this study.

ORCID ID

-- Please provide the ORCID for both corresponding authors. You should already received an email to request that

Order of manuscript parts

-- Title suppl. figure legends as such
-- please provide table legends (if needed) following the supplemental figure legends
-- please provide each table as a separate editable file, and provide a Table legend following the supplemental legends

Text changes

-- In the figure legends and supplemental figure legends - please bold panel labels eg. (A), (B)...
-- please remove supplemental figure legends from underneath each supplemental figure
-- please move supplemental references to the main reference list - format them as 10 authors et al
-- please provide each supplemental figure as a separate file (1 suppl figure per page)
-- in Supplemental figure S4 - there is a label (A) and (A) is mentioned in the S4 legend as well, but there is no (B) or (C) - please remove the sub-label

A. FINAL FILES:

B. MANUSCRIPT ORGANIZATION AND FORMATTING:

****Reviews, decision letters, and point-by-point responses associated with peer-review at Life**

Science Alliance will be published online, alongside the manuscript. If you do want to opt out of having the reviewer reports and your point-by-point responses displayed, please let us know immediately.**

Sincerely,

Shachi Bhatt
Executive Editor
Life Science Alliance

September 23, 2020

RE: Life Science Alliance Manuscript #LSA-2020-00827-TRR

Prof. Eviatar Nevo
University of Haifa+Institute of Evolution
199 Aba Khoushy Ave., mount carmel
Haifa 3498838
Israel

Dear Dr. Nevo,

Thank you for submitting your Research Article entitled "Incipient sympatric speciation in wild barley caused by geological-edaphic divergence". It is a pleasure to let you know that your manuscript is now accepted for publication in Life Science Alliance. Congratulations on this interesting work.

DISTRIBUTION OF MATERIALS:

Again, congratulations on a very nice paper. I hope you found the review process to be constructive and are pleased with how the manuscript was handled editorially. We look forward to future exciting submissions from your lab.

Sincerely,

Shachi Bhatt, Ph.D.
Executive Editor
Life Science Alliance